# Comparative Characterization of Human Meibomian Glands, Free Sebaceous Glands, and Hair-Associated Sebaceous Glands Based on Biomarkers, Analysis of Secretion Composition, and Gland Morphology

**DOI:** 10.3390/ijms25063109

**Published:** 2024-03-07

**Authors:** Yuqiuhe Liu, Igor A. Butovich, Fabian Garreis, Ingrid Zahn, Michael Scholz, Simone Gaffling, Samir Jabari, Jana Dietrich, Friedrich Paulsen

**Affiliations:** 1Institute of Functional and Clinical Anatomy, Friedrich-Alexander-Universität Erlangen-Nürnberg, 91054 Erlangen, Germany; lotus0807@hotmail.com (Y.L.); michael.scholz@fau.de (M.S.);; 2Department of Ophthalmology, University of Texas Southwestern Medical Center, Dallas, TX 75390, USA; 3Chimaera GmbH, 91058 Erlangen, Germany; 4Institute of Neuropathology, University Hospital Erlangen, Friedrich-Alexander-Universität Erlangen-Nürnberg, 91054 Erlangen, Germany

**Keywords:** meibomian glands, sebaceous glands, meibomian gland dysfunction, dry eye disease, cytokeratins, stem cell markers, cell–cell contacts, meibum, sebum

## Abstract

Meibomian gland dysfunction (MGD) is one of the main causes of dry eye disease. To better understand the physiological functions of human meibomian glands (MGs), the present study compared MGs with free sebaceous glands (SGs) and hair-associated SGs of humans using morphological, immunohistochemical, and liquid chromatography—mass spectrometry (LCMS)-based lipidomic approaches. Eyelids with MGs, nostrils, lips, and external auditory canals with free SGs, and scalp with hair-associated SGs of body donors were probed with antibodies against cytokeratins (CK) 1, 8, 10, and 14, stem cell markers keratin 15 and N-cadherin, cell–cell contact markers desmoglein 1 (Dsg1), desmocollin 3 (Dsc3), desmoplakin (Dp), plakoglobin (Pg), and E-cadherin, and the tight junction protein claudin 5. In addition, Oil Red O staining (ORO) was performed in cryosections. Secretions of MGs as well as of SGs of nostrils, external auditory canals, and scalps were collected from healthy volunteers, analyzed by LCMS, and the data were processed using various multivariate statistical analysis approaches. Serial sections of MGs, free SGs, and hair-associated SGs were 3D reconstructed and compared. CK1 was expressed differently in hair-associated SGs than in MGs and other free SGs. The expression levels of CK8, CK10, and CK14 in MGs were different from those in hair-associated SGs and other free SGs. KRT15 was expressed differently in hair-associated SGs, whereas N-cadherin was expressed equally in all types of glands. The cell–cell contact markers Dsg1, Dp, Dsc3, Pg, and E-cadherin revealed no differences. ORO staining showed that lipids in MGs were more highly dispersed and had larger lipid droplets than lipids in other free SGs. Hair-associated SGs had a smaller number of lipid droplets. LCMS revealed that the lipid composition of meibum was distinctively different from that of the sebum of the nostrils, external auditory canals, and scalp. The 3D reconstructions of the different glands revealed different morphologies of the SGs compared with MGs which are by far the largest type of glands. In humans, MGs differ in their morphology and secretory composition and show major differences from free and hair-associated SGs. The composition of meibum differs significantly from that of sebum from free SGs and from hair-associated SGs. Therefore, the MG can be considered as a highly specialized type of holocrine gland that exhibits all the histological characteristics of SGs, but is significantly different from them in terms of morphology and lipid composition.

## 1. Introduction

Dry eye disease (DED) is a condition of eye discomfort, itching, tiredness, burning, photophobia, redness, pain, and visual disturbances [1]. DED has received a lot of attention in recent years due to a large number of patients presenting with it and the fact that its symptoms are very disruptive to daily life. It is usually accompanied by an increase in tear film osmolarity [2,3]. DED can be divided into aqueous-deficient dry eye disease (ADDE) and evaporative dry eye disease (EDE) or a mixed type of both according to its pathophysiology [3,4]. EDE accounts for the majority of DED, and the most common reason for excessive tear evaporation is meibomian gland dysfunction (MGD) [3,5,6]. Defined at the 2011 International Workshop on MGD, it is a chronic, diffuse abnormality of the meibomian glands (MGs), usually characterized by obstruction of the terminal ducts and/or qualitative/quantitative changes in glandular secretion [5]. MGs are exocrine glands located in the tarsal plate of the upper and lower eyelids of mammals [7]. They are sebaceous glands (SGs) composed of a cluster of secretory acini arranged in a circular pattern around a long straight central duct and connected to it by smaller ductules [7,8]. The acini comprise meibocytes and follow a holocrine secretion mechanism. Basal meibocytes located in the periphery of the glandular acini develop via differentiating and mature meibocytes into hypermature meibocytes. These hypermature meibocytes, along with the lipids stored in them and formed during the ripening, produce an oily secretion. This secretion, called meibum, is transported via small excretory ductules into the main excretory duct and is finally released onto the tear film at the lid margin during lid closure [9,10]. During maturation, cells are distinguished by the increase in cell size, the increase in lipid droplet size, and the apoptosis of the nucleus [11]. 

SGs are widespread and abundant in the skin of the human face, scalp, and most skin areas except the palms and soles. Two types of SGs are distinguished as follows: one type is associated with a hair follicle (hair–sebaceous gland unit, to which a smooth muscle, musculus arrector pili, is also assigned), the so-called hair-associated SG; the other type is free SGs with no relationship to hair follicles. Such “free” SGs occur on the lips, nostrils (nasal wings), external auditory canals, nipples, and external genitals (labia minora) [12,13]. The structure of “free” SGs is similar to MGs; they consist of a single acinus or a bunch of acini that connect to the central duct by short ductules and a terminal end of the central duct that opens directly onto the skin’s surface. SGs’ acini are full of sebocytes; those cells show to a great part the same differentiation process as meibocytes in MGs [14,15]. 

Sebum is the secretion of SGs while meibum is the secretion of MGs; they both play a vital role in maintaining basic human biological activities [16,17,18,19]. Meibum seals the opposing lid margins during sleep, stabilizes the tear film, maintains a smooth optical surface for the cornea at the air–lipid interface, helps to spread the tear film on the ocular surface, reduces tear film evaporation, prevents spillover of tears from the lid margin, and prevents contamination of the tear film by sebum [14,16,19,20]. Sebum lubricates the skin and hair, makes them supple, and contributes significantly to the regulation of body temperature. In addition, the secreted sebum protects the skin, creating the physiologically acidic skin environment, thus conditioning the skin flora and enabling the skin to be protected against pathogens [18,20]. To date, there are only a few studies that have described the localization of human eyelid biomarkers in detail by immunohistochemistry [21,22]. However, there is no detailed study on the comparative location of biomarkers for free SGs or hair-associated SGs. Furthermore, despite the fact that meibum and sebum share certain compositional elements [23,24,25,26], recent pilot experiments with humans and mice demonstrated that the two secretions are different enough to justify their more detailed analysis and establish the relationship between the secretory products of the free SGs and the MGs. In order to achieve these goals, we compared MGs and three different types of free SGs in humans (specifically, “free” SGs of the lips, nostrils, and external auditory canal), as well as with hair-associated SGs of the skin, morphologically, by means of antibodies and with regard to their lipid composition. The central questions we wanted to answer were as follows: (1) Do MGs and free SGs differ structurally from each other? (2) Are there similarities and/or differences in the keratin, stem/progenitor cell markers, and desmosomes between these glands? (3) Is the appearance of desmosomes, as observed in MGs, a general mechanism of all free SGs and also of hair-associated SGs? (4) Are there differences in the lipid composition of the individual free SGs and can specific lipid fractions be derived from this that are of particular importance in the MG?

By addressing these questions, we intended to contribute to the general understanding of the similarities and differences between MGs and SGs in order to gain a basis for a deeper understanding of the evaporative form of DED and MGD, in addition to meibomian gland physiology, and provide new clues for the future treatment of DED/MGD.

## 2. Results

### 2.1. Histological Analyses of the Morphology of MGs, Free SGs, and Hair-Associated SGs

The structure of the MG (Figure 1A–C) shows that the main excretory duct is surrounded by many secretory acini. The acini are connected to the main excretory duct by short ductules. On the magnified image of the single acinus of the MG (Figure 1C), the acinus is filled with meibocytes. The basal cells in the periphery are surrounded by a basement membrane and differentiate from the outside toward the center of the acinus into mature and eventually overmature cells. As meibocytes differentiate from basal cells to overripe cells, the nuclei stained with hematoxylin gradually become darker and smaller (nuclear pyknosis); the cytoplasm stained with eosin gradually becomes lighter/foamier due to the incorporation of lipids. The histological structure of free SGs is similar to that of MGs where there are multiple acini surrounding a central duct, both connected by ductules to a main excretory duct (Figure 1D–L). Again, the holocrine secretion of the free SG is reflected in the structure of the acinar cells; the basal cells are located at the margin and the mature and overripe cells are in the center of the acinus. The changes in the nucleus and cytoplasm were similar to those seen in MGs. In the magnified image of the single acinus of a free SG (Figure 1F,I,L), the appearance of the basal cells transitions from a bluish staining with larger nuclei and a pink staining of the cytoplasm to a dark blue staining with smaller nuclei and a light pink staining of the cytoplasm in the mature and hypermature cells. The accumulation of lipid droplets results in enlargement of the cells and a pale appearance. Also, the histological structure of hair-associated SGs includes the above-described acini, connecting ducts, and a main central excretory duct that opens at the base of the hair funnel (infundibulum) (Figure 1M–O). Several acini are located on one side at the top of the hair follicle. The cell maturation process of hair-associated SGs is similar to those of MGs and other SGs. Basal cells are located at the peripheral edge of the acini. Toward the center, increasingly differentiating, then mature, and finally hypermature sebocytes are seen. The nuclei become pyknotic centrally, with the cytoplasm being foamy (Figure 1N,O).

### 2.2. Immunohistochemical Localization of Biomarkers in MGs, Free SGs, and Hair-Associated SGs

Since a large number of antibodies were tested on the various glands during the study, only the results for cytokeratin 10 (CK10) and keratin 15 (KRT 15) are shown here as examples. All other figures can be found in the Appendix A. The results of the immunohistochemical investigations are summarized in Table 1. For better comparability, only reaction results obtained from one and the same body donor are visually depicted here. However, they were comparable in the investigated samples from different body donors.

#### 2.2.1. Cytokeratins

CK1: Immunohistochemical detection of CK1 showed differential expression in MGs, free SGs, and hair-associated SGs (Table 1). CK1 was strongly expressed in the superficial layer of the central duct of MGs and free SGs (Appendix A), but not in the basal layer. Both MG and free SG acini showed no or very weak staining for CK1 as well. Hair-associated SGs from the scalp revealed different staining than the MGs and free SGs. Here, CK1 was expressed not only in the superficial layer of the central duct but also in the acini (Appendix A). 

CK8: CK8 was expressed in MGs, free SGs, and hair-associated SGs (Table 1). CK8 was slightly differentially expressed as the MGs showed positive reactivity in the basal cells and connecting ductules (Appendix A), whereas in free SGs and hair-associated SGs, it was only expressed in the basal cells but not in the connecting ductules (Appendix A).

CK10: CK10 was expressed in the superficial layer of the central duct but not in the MG acini (Figure 2A,B) (Table 1). Free SGs from the lip, nasal wing, external auditory canal, and scalp tissue showed similar expression levels (Figure 2D,E,G,H,J,K,M,N). Here, CK10 was expressed in the superficial layers of the central duct and in the SG acini. However, staining in the hair-associated SG acini was more intense than the staining of the free SG acini (Figure 2M,N). 

CK14: The expression of CK14 in the free SGs showed a differential expression compared to MGs and hair-associated SGs (Table 1). The expression of CK14 could be detected mainly in the basal cells and the differentiating cells of the MGs and hair-associated SGs (Appendix A). Additionally, in free SGs, the mature cells were reactive (Appendix A). It was also expressed in the basal layer of the central duct of all three gland types (Appendix A).

#### 2.2.2. Stem Cell Marker

KRT15: Reactivity of KRT15 was visible in the epithelial lining of the connecting ductules (the transition zone) of the MGs (Figure 3A,B) (Table 1). In free SGs, KRT15 was expressed in the transition zone and basal layer of the central duct (Figure 3D,E,G,H,J,K). In addition to the transition zone and basal layer of the central duct in hair-associated SGs, reactivity was also detected in the basal cells of the acini (Figure 3M,N). 

N-cadherin: The antibody against N-cadherin did not show positive reactivity in two of the body donor samples. In the human hair-associated SGs, N-cadherin was expressed on the basal cells and the main excretory duct (Appendix A; Table 1). In the MGs, nasal wing, lip, and external auditory canal, N-cadherin was only expressed on the basal cells of the acini (Appendix A; Table 2).

#### 2.2.3. Cell–Cell Contact Marker

Desmoglein 1 (Dsg1): Dsg1 was expressed somewhat differently in paraffin sections of the four analyzed body donors. In one of the four donors, Dsg1 reacted more intensely in the basal cells and only weakly in the mature cells of the MG (Appendix A). In the other three samples of body donors, Dsg1 was more equally distributed over the entire MG acinus (Appendix A). All four tissues of the human body donors had in common the Dsg1 reactivity on the epithelial cells of the central duct. The expression pattern for Dsg1 in free SGs from the nasal wings, lips, and external auditory canals, and in hair-associated SGs from the scalp was comparable. Here, it was expressed on the excretory duct epithelial cells, basal cells, differentiating cells, and mature cells (Appendix A). 

Desmoplakin (Dp), Desmocollin 3 (Dsc3), Placoglobin (Pg), and E-cadherin: Dp, Dsc3, Pg, and E-cadherin were expressed similarly in MGs, free SGs, and hair-associated SGs (Appendix A) (Table 1). They were expressed equally on the excretory duct epithelial cells, basal cells, differentiated cells, and mature cells of MGs, free SGs, and hair-associated SGs (Appendix A). Claudin 5: Claudin 5 as a tight junction protein did not react on MGs, free SGs, or hair-associated SGs but reacted only with the endothelial cells of blood vessels (Appendix A) although the positive control worked fine (Appendix A). 

### 2.3. Lipid Accumulation in MGs, Free SGs, and Hair-Associated SGs

In the eyelids, lipid droplet accumulation could be observed in the MG’s central duct and inside the acini. The size of the lipid droplets enlarged during the differentiation. The mature cells were filled with lipid droplets, while in the basal and differentiating cells, the lipid droplets did not fill the whole cell (Figure 4A–C). The amount of secretion in the free SG central duct was visible but not as high as that in the MG central duct. In the free SG central duct, lipid droplets were not filled with the entire lumen. In free SGs from nasal wings (Figure 4D–F), lips (Figure 4G–I), and external auditory canals (Figure 4J–L), the lipid droplets were observed in basal, differentiating, and mature cells. However, the area in the mature cell was larger than in the basal and differentiating cells. Lipid droplets in hair-associated SGs (Figure 4M–O) showed similar morphology to free SGs from the nasal wings, lips, and external auditory canals. Lipid droplet size was slightly larger in mature cells than in basal cells without an accumulation of secretion in the excretory duct.

### 2.4. Meibum and Sebum Component Analysis

#### 2.4.1. Basic Information about the Volunteers

In total, there were fifteen volunteers enrolled in this study, five males and ten females. A *Schirmer I* test was performed to exclude participants with DED symptoms. Of the initial fifteen volunteers, one female had less than 10 mm secretion length and thus was excluded from collecting study samples. Thus, a total of 14 secretions were collected from volunteers aged between 22 and 66 years old (Median age = 49.5 years old). All volunteers denied having diseases or inflammation of the following sampling areas: eyelids, nasal wings, external auditory canals, and scalps, with no dry eye symptoms reported. 

#### 2.4.2. Comparison of the Composition of Meibum and Sebum

Due to the insufficient weight of the collected secretions from MGs and SGs of some of the donors, a total of 55 samples were analyzed. Note that because of the space constraints, the results of the detailed lipidomic analysis of these secretions are to be reported in a separate publication. 

Initially, the raw LCMS files were imported in the Progenesis QI software (from Waters Corp., v2.3), aligned, and analyzed using its built-in PCA algorithm. The following parameters of the PLS-DA model were used: Pareto scaling with no transformations; four components; Variance explained–R2Y(Cum) 72%; Variance predicted–Q2(Cum) 61%. There were 410 variables (i.e., possible lipid analytes) with LCMS abundances of ≥0.01% of the base peak found by the Progenesis software; therefore, the software analyzed a 55 × 410 dataset for four groups (meibum from eyelid, sebum from nasal wings, external auditory canals, and scalps). Seven variables were automatically excluded by the software from further processing, resulting in 403 final analytes. Then, the pre-processed data sets were exported in the EZinfo software (v3.0.3.; from Umetrics, a part of Progenesis QI software package) and subjected to a more detailed analysis using its PCA and PLS-DA routines.

First, the data were plotted as a PCA plot (Figure 5A). This unbiased, unsupervised approach demonstrated an extremely tight group of samples (scores) S1, a rather tight group of samples S2, and a group of broadly spread samples S3. When the scores were color-coded based on their respective types of secretions (Figure 5B), it became evident that the group S1 was produced by meibum samples, group S2 was a combination of overlapping sebum samples from nasal areas, scalps, and external auditory canals, while group S3 was composed solely of the sebum samples collected from the external auditory canals. To further illuminate the intergroup differences, the data were processed using the PLS-DA template of EZinfo. Notably, the two approaches—an unbiased, unsupervised PCA and supervised OPLS-DA—resulted in almost identical grouping of the scores (study samples). 

The close similarity between the PCA and the OPLS-DA data allowed us to proceed with the next steps of the data analysis using the latter approach, by conducting a preliminary analysis of the possible lipid discriminants that were responsible for the separation of meibum and sebum samples using the OPLS-DA approach. Importantly, the OPLS-DA Loadings Bi-Plot (Figure 5C), which simultaneously shows the scores and the variables (i.e., lipid analytes with unique combinations of *m*/*z* values and LC retention times), revealed variables which were associated with specific groups of samples. 

However, the Loadings B-Plot does not provide information on the statistical significance of the experimental observations. To obtain this information, the data were arranged in the Variable Importance in Projection (VIP) plot (Figure 5D). The 25 most influential variables were positively identified as cholesteryl esters (signals with an *m*/*z* value of 369.3523), wax esters (e.g., *m*/*z* 647.6696 and 619.6381), and triacylglycerols (mostly visible as a range of diacylglycerol fragments formed in-source due to a neutral loss of one of the fatty acid fragments). Some of the study samples had noticeable amounts of chemical contaminants, for example an oxidized form of a plasticizer Irgafos 168 with an *m*/*z* value of 663.4535. These analytes/signals were excluded from further analyses.

Finally, the data were arranged as the Hotelling’s T^2^ Range plot (Figure 5E), from which only one sample of ear wax was found to exceed the T^2^ Crit (99%) and could be considered as an outlier. All meibum samples (orange circles) produced a tight group of samples with low inter-sample, intra-group variability, while sebum samples from external auditory canals (green circles), nasal wings (blue circles), and scalps (red circles) produced much looser groups. Notably, sebum from external auditory canals demonstrated the widest spread of the samples which was an indicator of its highest variability, while the other two types of sebum appeared to be quite similar to each other in that respect.

### 2.5. Three-Dimensional Gland Reconstruction of Different Glands

Using volumetric tissue modeling, a three-dimensional image of the corresponding tissue sample/tissue block was generated via the HiD software (v1.0), from which the respective structure of interest to the viewer could be isolated by annotation (Appendix A). This was carried out for all SGs and MGs to be examined, so that at the end, three-dimensional images of the glands were available for comparison of the glands with each other (Figure 6).

The comparative study between the different glandular localizations showed that the size of the glandular acini hardly varied between the SGs. However, the number of glandular acini and the size of the glands varied considerably. For example, the MGs in the upper eyelid had between 300 and 500 glandular acini, and in the lower eyelid, between 250 and 350. All acini opened very regularly into the central excretory duct via smaller connecting ductules (Figure 6A). In the external auditory canal, the glandular acini of the SGs were not as sharply demarcated as in the other glands and tended to “fuse” together. They were beneath the papillary dermis (Figure 6B), which they penetrated with their excretory duct. The glands were asymmetrically shaped and the excretory duct broke through the epithelium in an oblique course. The number of glandular acini here varied between 10 and 30 in both female and male samples. Hair-associated SGs had the smallest number of glandular acini (5–10 per gland). They had one or two excretory ducts that opened at the base of the hair funnel (infundibulum) (Figure 6C). SGs in the lip region had a single excretory duct into which smaller connecting ductules opened. The duct system branched like a deciduous tree (Figure 6D,F). At the lower part of the tree trunk, rarely single glandular acini opened. The glands varied in size with 20–40 glandular acini with no significant differences between females and males (Figure 6D,F). In the area of the nostril, SGs were present, which were significantly larger in men than in women. The main excretory duct was often branched (Figure 6E), and smaller connecting ducts were located in the main excretory ducts. The glandular acini extended far into the epidermis. The number of glandular acini per gland was between 50 and 80 in males and 40 and 60 in females.

## 3. Discussion

Recently, Verma et al. summarized the major events in MG development in a concise article [14]. They state that MGs are modified SGs and as such both follow a holocrine secretion mechanism and share various common developmental features [14]. As skin appendages, the early development of MGs is very similar to the early development of hair follicles and SGs [27]. The authors stated that SGs and hair follicles have been studied much more intensively than MGs to date, and that by understanding the differences and similarities between MGs and SGs, one can draw conclusions about the normal physiology of MGs and their changes over time, in order to draw therapeutic approaches for the treatment of MG-associated diseases including MGD. This is where our study contributes by shedding further light on the similarities and differences between MGs and SGs. In the International Workshop on Meibomian Gland Dysfunction: Report of the Subcommittee on Anatomy, Physiology, and Pathophysiology of the Meibomian Gland [7], the dimensions and number of MGs in the upper eyelid and lower eyelid for humans are summarized. Verma et al. state in their overview table for the differences between MGs and SGs that SGs are on average smaller than MGs [14]. Our three-dimensional reconstructions support this finding and furthermore show that not only the shape of MGs differs from other free and hair-associated SGs, but also that the shape of the different SGs differs among themselves. For example, SGs in the area of the nostrils often do not have just one main excretory duct, but the ductal system is displayed (Figure 6E). Free SGs in the area of the lips have a rather plump shape compared to MGs or SGs in the area of the nostrils. Furthermore, our findings show that the number of secretory acini in MGs (especially of the upper eyelid) is significantly higher (300–500) than in the compared SGs and that this number is many times higher than the number of about 10–15 per gland cited in [7] related to [28]. The number of secretory acini of MGs in the lower eyelids is almost comparable to the number of large SGs on the nostril. We can only speculate here that the differences in the development between MGs and SGs as summarized in the review [14] contribute to the different morphology. However, it is interesting to note that the size of the gland(s) does not seem to influence the composition of secretions, as our findings show that the glandular secretion of the SGs of the nostril and scalp is very similar, whereas it differs from the secretion in the external auditory canal and MGs (see below).

Our present results also show that there are both similarities and differences between MGs and SGs with regard to various biomarkers [cytokeratins: CK1, CK8, CK10, CK14, the stem cell markers: keratin15 and N-cadherin, and the cell–cell contact markers: Dsg1, Dsc3, Dp, Pg, E-cadherin, and claudin 5 for the tight junctions (TJ)] that have only been analyzed in a few studies to date and have involved MGs to some extent [14,21]. In this context, it is yet unknown whether MGs, free SGs, and hair-associated SGs of other localizations are comparable and could possibly be replaced by other free or hair-associated SGs if required. Cytokeratins are proteins of the intermediate filaments of the cytoskeleton of epithelial cells. Their main function is the maintenance of cell morphology, tissue stability, and cell–cell communication [29]. They can be used as biomarkers to identify the subtype and differentiation status of epithelial cells [30]. Previous studies investigating the expression of cytokeratins in MGs revealed that CK1 and CK10 were expressed in the main execution ducts of MGs [22,31,32,33]. These findings were confirmed in the current study. In an early study on human SGs, CK10 was found to be strongly expressed in immature sebocytes (basal cells) and cells of the SG excretory duct, but only weakly expressed in mature SGs [34]. No previous study has described the expression of CK1 and CK10 in free SGs. In our study, CK1 expression in free SGs was the same as in MGs. In contrast, CK10 was expressed not only in the superficial layer of the excretory duct but also in SG acini. In hair-associated SGs, CK1 and CK10 expressions showed the same pattern in the superficial layer of the excretory duct and in the acini of hair-associated SGs. The pattern of CK1 and CK10 expression in SGs suggests keratinization of the epithelium of the central excretory duct. CK10 may thus be a useful marker for the diagnosis of MGD, as the keratinization of normally non-keratinized epithelium often shows abnormal reactivity of CK10 [22,35]. The reactivity of CK8, which is known to be mainly expressed in secretory epithelial cells of the acini [36], was detected in the basal layer of the MG central duct and in the basal cells of the acini in the present study. However, in a previous study, CK8 was found to be expressed during all differentiation stages of MG acinar cells in humans [22]. In free SGs and hair-associated SGs, CK8 was expressed only in the basal cells of the acini, but not in the basal layer of the central excretory duct. This is in contrast with the results of a previous study in which no expression of CK8 was detected in the acini of SGs [37,38]. CK14 was expressed on the basal and differentiating cells of the MG acini and the basal cell layer of the main excretory duct and ductules, which is consistent with a previous study [39]. In this study by Liu et al., conducted in monkeys, only the basal cells of the MGs expressed CK14 [31]. However, in other studies on human samples, CK14 was also detected in the acini of MGs [22,35]. However, the two cited papers do not describe in detail to which differentiation stage the CK14 expression in meibocytes refers, but only say that it was detectable in meibocytes. Moreover, it is clear from the published figures in these two publications that CK14 was expressed by both basal cells and differentiating cells [35]. Our results also show CK14 on mature sebocytes of free SGs and hair-associated SGs, which is also consistent with other studies [37,38]. Interestingly, CK14 is mainly expressed by basal cells in MGs. In contrast, it is expressed by basal cells, mature cells of free SGs, and hair-associated SGs. In the present study, the expressions of CK8, CK10, and CK14 in MGs differ from those of free SGs and hair-associated SGs. Obstruction of the MG due to hyperkeratinization of the epithelial cells of the main duct is the most common pathological finding observed in MGD [7]. Alterations in cytokeratins can lead to hyperkeratosis [40]. This suggests that CK10 is a useful marker for the diagnosis of MGD, as keratinization of non-keratinized areas often shows abnormal reactivity of CK10. Interestingly, CK14 was mainly expressed by the basal cells of the MGs. In contrast, it was expressed in basal cells, mature cells of free SGs, and hair-associated SGs. This suggests that cell differentiation may be different in MGs, free SGs, and hair-associated SGs. 

Stem cells can differentiate into specialized cell types [41]. Since MGs are histologically similar to, but morphologically and biochemically different from, SGs, it remains unclear whether the expression of stem/progenitor cell markers is the same in MGs, free SGs, and hair-associated SGs. Different views have been proposed regarding the existence and location of stem/progenitor cells in MGs [9,22,39,42] which has led to numerous controversies. Some authors consider that stem cells are found in the periphery of the MG acini [9,43], while others consider that these cells are located in the transitional area between the MG ductule and the acinus [39,42] (summarized in [44]). In the current study, only the MG ductules showed KRT15 reactivity, while in free and hair-associated SGs, KRT15 was expressed in both the ductules and the central duct of free and hair-associated SGs. The expression of KRT15 on MGs was verified in the study by Tektaş et al. [22]. The expression of KRT15 has been described as a stem cell marker in the bulge of hair follicles [45], which was also found in the present study. None of the previous studies investigated the expression of KRT15 in free SGs. In a recent study by Parfitt et al. [39], it was shown that marker-retaining cells were located in the transition zone between the ductules and the acinus and that these marker-retaining cells were probably stem cells. This suggests that the MG stem cells may be localized in the epithelial cells of the transition zone between the acini and the ductules. However, it is less clear whether the location of the stem cells changes with age and sex or whether stem cell expression is the same in MGD patients. The concentration of stem cells in this region might be related to embryonic development. Further studies should focus on patients with DED or different age groups to see if the stem cell markers are related to these variables.

In the MG acini, the basal cells are located between the basal lamina and the differentiated cells [10]. These cells are connected to the basal lamina by hemidesmosomes [11,46]. The differentiated cells are connected to the neighboring basal cells by desmosomes [11]. Previous studies by Rötzer et al. have shown that desmosomes are present in all layers of the MG and are especially abundant in the mature cells [21]. Further detailed studies on desmosome expression in the MGs are currently not available. Dp, the most important component of desmosomes, was expressed only in the mature cells [21,47]. The expression of these desmosomes in other holocrine-secreting glands, such as free SGs and hair-associated SGs, has not been described previously. In our study, desmosomes were evenly distributed between basal, differentiating, and mature acinar cells in MGs, free SGs, and hair-associated SGs, and there was no significant difference in desmosome expression between MGs, free SGs, and hair-associated SGs. Moreover, they were equally expressed in all layers of the central duct and ductules of MGs, free SGs, and hair-associated SGs. To clarify whether the observed pattern was restricted to desmosomes or typical for cell–cell contacts in MGs and SGs, E-cadherin and claudin 5 were additionally investigated. E-cadherin is an adhesion molecule for the Adherens Junction (AJ) and claudin 5 is a marker for the TJ. E-cadherin was equally expressed by basal, differentiating, and mature acinar cells in MGs, free SGs, and hair-associated SGs. All three types of glands showed no expression of claudin 5. These expressions were demonstrated in a previous study [21]. The expression of Dp in our study differed from previous studies [21,48] in which it was expressed only in the mature acinar cells of MGs. In the present study, it was found in both basal and mature acinar cells. The reasons for this discrepancy can be explained as follows: the number of donor samples from the human body was probably too small, and the experimental conditions and tissues were different, which ultimately led to different results for Dp expression in this experiment than in the previous study. Since Dp was mainly expressed in mature cells of the MG, the authors discuss that the distribution pattern of the components of desmosomes depends on the differentiation status of the meibocytes, with more desmosomes occurring in mature cells of the MGs [21,48]. The results presented here suggest that in human tissue desmosomes, AJ and TJ are not differentially expressed during acinar cell differentiation. The acinar cells of MGs, free SGs, and hair-associated SGs showed the same reactivity of Dsg 1, Dsc3, Pg, Dp, and E-cad, and this occurrence may be a general feature in the three gland types.

Finally, our study also provides further insight into the comparison of lipid composition between MGs and SGs. Previously, clear differences in the lipid composition between SGs and MGs of both humans and mice have been found and discussed [14,26,49,50,51,52,53]. However, our current study is the first to compare glandular secretions from MGs and SGs of three different types from the same donors. As shown by ORO staining, the size of the lipid droplets was larger in the mature cells than in the basal acinar cells. This suggests that overall lipid production is related to the differentiation state, which is consistent with other studies in humans and mice in which mature MGs and meibocytes were shown to synthesize and accumulate meibum [21,54,55]. Though the initial comparison of human meibum and sebum from nasal and forehead areas has been conducted, no study has yet examined the lipid distribution in human SGs of various types. The central ducts of MGs were filled with lipids, whereas the central ducts of free SGs and hair-associated SGs had less lipids compared to MGs, which could mean that the secretions of these two types of SGs contain less lipids and, possibly, are enriched with other substances besides lipids. This idea was also confirmed in the subsequent data analysis study. LC-MS analysis revealed that the meibum component was very different from the sebum of the nostrils, external auditory canals, and scalp. Wax esters, cholesterol esters, free fatty acids, free cholesterol, and glycerides were currently found in both human sebum and meibum [23,25,26,56]. This finding was also confirmed in our study. In addition, meibum contains (O)-acylated ω-hydroxy fatty acid (OAHFA), ceramides, cholesteryl esters of OAHFA, diesters, triacylglycerides (TAG), free cholesterol, free fatty acids, and phospholipids, a complex mixture of polar and nonpolar lipids [26,56]. The differences in composition between meibum and sebum are that sebum contains higher molar amounts of triglycerides, squalene, diglycerides [23] (summarized in [14]), and shorter chain wax esters [49,53]. Some studies have found changes in the composition of meibum in MGD patients with MG excretory duct blockade [57,58,59]. The current study showed that the components of meibum and sebum from free SGs and hair-associated SGs were very different from each other. On the contrary, the composition of sebum from the nostrils and scalp were quite similar, while the sebum from the external auditory canal was close to them but still different. This could be due to the fact that the sebum from the external auditory canal was secreted by both SGs and ceruminous glands [60]. However, the exact lipid composition of the ceruminous gland secretions is yet unknown and should be examined in future studies. 

LC-MS analyses revealed that the study samples contained exogenous impurities such as common plasticizers Irgafos 168 and fatty acid amides, such as oleamide (not mentioned in the Results). The plasticizers are typical impurities that originate either from organic solvents or can be leached from plastic products. The glass vials used in this study had Teflon-coated lids, and the stainless steel spatula and MG compressor did not come into contact with the collector’s fingers or other sources of contaminants during sample collection. The occasional presence of contaminants may have been due to the repeated opening and closing of the lid; the laboratory’s plastic gloves may have touched the lid or the opening of the glass vial, resulting in contamination. In addition, the reason for detecting contaminants may also be due to previous ingestion, ingestion through food or the eyes, or wearing contact lenses (although the wearing of contact lenses was asked of the subjects and would have led to study exclusion if used). Currently, almost nothing is known about the pathways of nano- and microplastics and their effects on the human system, as shown in a recent review article on this topic. Future research should completely avoid any possible contact of secretions, collection tools, and collection containers with plastic products during sample collection and handling. The present study is limited by the fact that it only used tissue samples from four body donors and thus only covered a rather small “n” number. In addition, because we were only able to use just one female body donor, it is also not possible to make any real statements about sex differences, though no major differences in meibum lipid composition between males and females were found when studying human and mouse specimens. Future comprehensive, comparative proteomic and lipidomic analyses could provide datasets that provide a larger arsenal of biomarkers, proteins, and lipids of interest for understanding healthy and dysfunctional MGs. 

## 4. Materials and Methods

### 4.1. Collection of Human Samples

Eyelids, nostrils, lips, external auditory canals, and scalp tissues were collected from 4 body donors (3 males and 1 female, age range = 68–79 years) who have made a testamentary disposition for research and science to the Institute of Functional and Clinical Anatomy, Friedrich Alexander University Erlangen-Nuremberg (FAU), Erlangen, Germany. None of the body donors had recent trauma, infections (including the donor site), or other known diseases that may have affected the donor sites in question. The tissue was collected from the body donors within 24 h after death by qualified personnel of the Institute.

### 4.2. Fixation and Preparation of the Tissue 

After preparation, all tissues were immediately fixed overnight, but for no longer than 24 h, in 4% paraformaldehyde (PFA) in phosphate-buffered saline (PBS) at room temperature. The tissue samples were then rinsed in PBS, embedded in paraffin, and cut into 6 μm sections. Each of the 10 sections was stained for a morphological view with hematoxylin and eosin (H&E) according to a standard protocol and examined and photographed under a Keyence BZ-X810 all-in-one fluorescence microscope (Osaka, Japan).

### 4.3. Immunohistochemistry

Immunohistochemical investigations were analyzed on the tissue sections of the different localizations with various biomarkers that play a role in particular in the context of examinations of the MG [22,48,61,62]. These were antibodies against the cytokeratins (CK) 1, 8, 10, and 14, the stem cell markers keratin 15 and N-cadherin, as well as the cell–cell contact markers desmoglein (Dsg) 1, plakoglobin (Pg), desmoplakin (Dp), desmocollin (Dsc 3), and claudin 5. For this purpose, the paraffin sections were deparaffinized and rehydrated in a descending alcohol series. For antigen retrieval, the sections were washed twice in distilled water (for 5 min each). Afterwards, the sections were boiled for 20 min in citrate buffer (pH = 6.0) or steamed in glycine-EDTA buffer (pH = 9.0) according to the antibody manufacturer’s instructions (Table 2). Subsequently, the sections were cooled at room temperature for at least 1 h and washed in distilled water. Endogenous peroxidases were inactivated with 3% hydrogen peroxide for 20 min. Afterwards, the slides were washed twice for 5 min with distilled water. Non-specific binding was blocked with 10% normal goat serum (Dako AG, Wiesentheid, Germany) in TBST (Tris-buffered saline with Tween-20). The sections were then treated with an avidin/biotin blocking kit (BioLegend, San Diego, CA, USA) for 10 min before incubation overnight at 4 °C with the primary antibodies diluted in TBST (Table 2). This was followed by incubation with biotinylated secondary antibodies (Table 2) for 1 h at room temperature, followed by incubation with VECTASTAIN^®^ Elite ABC Kit (Vector Laboratories, Burlingame, CA, USA) for 1 h. The sections were then stained with AEC substrate solution (Dako AG, Wiesentheid, Germany) and counterstained with hematoxylin. A negative control, incubation without the primary antibody, was run on each section. The sections of the eyelids with MGs, in which all the tested antibodies had already been detected, served as positive controls [22,48,61,62] except for claudin 5. Here, the tissue section of the human kidney served as the positive control [63].

### 4.4. Oil Red O Staining

After 4% PFA fixation and washing, the tissue was dehydrated in 30% sucrose in PBS and embedded at −25 °C in O.C.T. compound (Tissue-Tek^®^, Torrance, CA, USA). Sagittal 10 μm thick sections were made from the tissue pieces. Oil Red O (ORO) stock solution was prepared by adding 300 mg of ORO powder (Chroma-Gesellschaft Schmid GmbH & Co, Stuttgart, Germany) to 100 mL of 99% (vol/vol) isopropanol and was mixed well at 37 °C. The cryosections were air dried at room temperature for at least 1 h, then washed in PBS for 10 min, and soaked in 60% isopropanol for 5 min. Sections were then stained with ORO working solution for 10 min (ORO stock solution was diluted 3:2 with distilled water) after filtering with 0.22 µm syringe filters (Carl Roth GmbH & Co. KG, Karlsruhe, Germany). The sections were placed in 60% isopropanol and rinsed until the solution was clear to completely remove the excess staining. The sections were then counterstained with hematoxylin.

### 4.5. Analysis of Secretion Composition 

#### 4.5.1. Collection of Meibum and Sebum

A total of 14 subjects (5 males and 9 females, age range = 22–68 years) who had no ocular disease or infection at the collection site (eyelids, nostrils, external auditory canals, and scalp) and underwent a Schirmer I test to rule out DED participated in this study. Samples collected from different subjects were analyzed individually. The subjects were recruited locally via the Institute of Functional and Clinical Anatomy at FAU. The study was approved by the Ethics Committee of the FAU (#84_19 B). All recruited subjects signed informed consent forms. The samples were collected for LCMS analyses as follows: Pre-weighed dry glass vials were filled with 1 mL of a chloroform/methanol solvent mixture of (2:1, vol/vol) LC. Meibum was expressed from the eyelids of the subjects using MG stainless steel tweezers (eyelid massage tweezers meibum expressor bought via Amazon.de) and collected from the eyelid margin at the MG openings with a stainless steel microspatula (also bought via Amazon.de). Similarly, sebum was collected from each subject with a stainless steel spatula from the nostril area (after its compression with cotton Q-tips), as well as from the external auditory canals and the scalp (i.e., a sterile cotton swab was used to perform a rough cleaning of the collection site. Then, the stainless steel removers were used to apply repeated firm top–down pressure to the site until secretions were seen. Lastly, all secretions were collected by the remover and transferred to glass vials with a solvent mixture). In order to avoid any inadvertent contamination with exogenous materials, care was taken to ensure that the secretions did not come into contact with the cotton swab or the fingers of the experimenter. The lipid samples thus obtained were transferred from the stainless steel spatula to the chloroform/methanol solvent mixture in the glass vials. The dissolution of the lipid at room temperature was facilitated by quick stirring of the spatula until the secretion had completely liquefied. The solvent was then completely evaporated in a stream of dry nitrogen using an evaporator (Appendix A) so that only the sample remained in the vial. The vials containing the samples were weighed again to determine the weight of the dry secretions. The samples thus obtained were stored at −80 °C until they were analyzed by LCMS (see below).

#### 4.5.2. Liquid Chromatography–Mass Spectrometry and Statistical Analysis

The secretion composition was analyzed by C_18_ reverse-phase LCMS in the positive ion mode using the atmospheric pressure chemical ionization (APCI) technique. A Synapt G2-Si high-resolution time-of-flight mass spectrometer equipped with an IonSabre II APCI ion source and connected to an Acquity M-class ultra-high pressure LC system was used (all from Waters Corp., Milford, MA, USA). The LCMS system was operated under the MassLynx software (v.4.0.; Waters Corp.), while the raw LCMS data were processed in the Progenesis QI software (version 2.3; also from Nonlinear Dynamics/Waters Corp.). Recently described protocols were used to analyze the secretions. All analyses were performed on four sample groups as follows: meibum from the eyelids, sebum from the nasal wings, external auditory canals, and scalps. After the initial evaluation of the secretions in the MassLynx software and pre-processing the raw data in the Progenesis QI software, the prepared data were imported into the EZinfo software (v3.0.3; from Umetrics/Waters Corp., Milford, MA, USA) and analyzed by its Principal Component Analysis (PCA) and Partial Least Squares Discriminant Analysis (PLS-DA) routines to obtain information on the possible relationships between the four sample groups. Several types of data plots were created. The PLS-DA scores plot was used to determine the differences, similarities, and groups of the secretions. The PLS-DA Bi-plot was used to assess the interrelationships of the secretions. Hotelling’s T^2^ test is a multivariate version of the Student’s test and provides a tolerance region or the desired confidence limit (usually 95% or 99%) for the data in a two-dimensional scores plot. The Variable Importance in Projection (VIP) plot is a measure of a variable’s importance in the PLS-DA model and is often used for variable selection.

### 4.6. Analysis of the Gland Morphology

#### 4.6.1. Serial Sectioning and Light Microscopy

Tissue blocks that contained MGs, free SGs in the area of the nose, lip, external auditory canal, and hair-associated SGs in the area of the scalp taken from body donors (3 males and 1 female, age range = 68–79 years) were cut out en bloc after fixation in 4% paraformaldehyde, dehydrated in graded concentrations of ethanol, and embedded in paraffin. Serial sections (6 µm) in a sagittal plane were stained with hematoxylin and eosin.

#### 4.6.2. Scanning of Serial Sections

The serial sections of the MGs, the free SGs in the area of the nose, the lip, the external auditory canal, and the hair-associated SGs in the area of the scalp were then scanned with a Hamamatsu NanoZoomer S60 Digital Slide Scanner (Hamamatsu, Japan). All digitized images were transferred and then processed using the HiD^®^ 3D application (Chimaera GmbH, Erlangen, Germany) to reconstruct and visualize the cut morphology of the gland types.

#### 4.6.3. Processing and 3D Reconstruction

Extensive processing was then performed to create three-dimensional models, some of which have already been described by the author group [64,65]. As described in [65], digitized histological images can potentially exhibit numerous artifacts such as nonlinear tissue deformations, heterogeneous intensities, and low contrast. These artifacts are not uncommon and are influenced by tissue-sectioning techniques, the type and duration of chemical exposure during staining, and environmental factors such as temperature. The first step is to undo these intensity artifacts using histogram-matching methods appropriate for image sequences, resulting in standardized intensity images. Then, the tissue slices in each individual slice sequence were aligned using automated software programs through a process called rigid registration. This step was important to overcome the different alignment of the tissue within the different slices. Subsequently, the nonlinear tissue deformations that resulted from cutting and further processing were reversed using nonrigid registration methods. These methods compute translation vectors for each image pixel to best match the corresponding image pixels of neighboring images. These corrected histological images were then processed, labeled, and superimposed to obtain a final volume image of the labeled structure or a three-dimensional image of the original tissue that can be visualized by volume rendering. All steps of the reconstruction process were realized in the three-dimensional reconstruction application HiD^®^ mentioned above. Finally, the spatial relationships were precisely completed by the annotation tool of the Chimaera AI-B2 client.

## 5. Conclusions

In conclusion, all our results support the view that MGs exhibit all the histological characteristics of SGs, but significantly differ from them in terms of morphology and lipid metabolism. The present study provides a basic initial comparison of the histological characterization of different types of biomarkers and secretory components in MGs, free SGs, and hair-associated SGs. Further studies, especially in humans with regard to development, maturation, and aging, as well as sex differences, may provide further information on the characterization of MGs, free SGs, and hair-associated SGs. The present study lays the foundation for further investigations, firstly with regard to the comparison of the findings obtained in “healthy” individuals with dysfunctional meibomian glands as in DED cases to examine the differential expression of biomarkers and secretion composition. In addition, a comprehensive lipidomic analysis of the comparison of the secretion profiles of the different types of sebaceous glands and meibomian glands will follow.

## Figures and Tables

**Figure 1 ijms-25-03109-f001:**
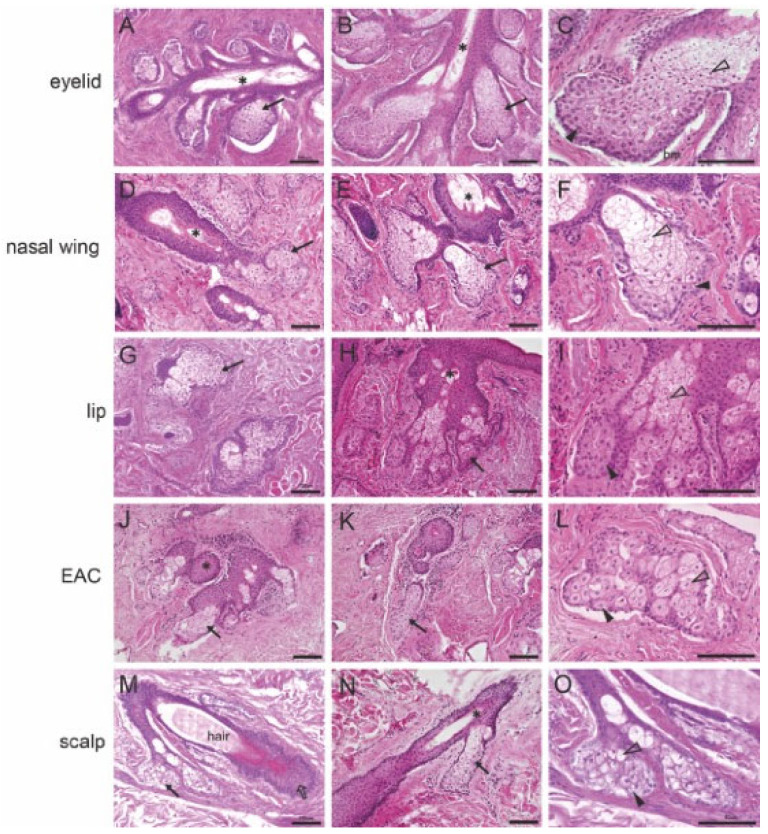
Histological analyses of the morphology of human meibomian glands (MGs), free sebaceous glands (SGs), and hair-associated SGs. (**A**,**B**) H&E staining of the human MGs. The MGs comprised several acini (arrow), a central duct (asterisk), and connecting ductules that connect the acinus and the central duct. (**C**) Human MG acini. Basal cells (arrowhead) were located at the peripheral margin of the acinus and attached to the basement membrane (bm). Mature cells (open arrowhead) were located in the center of the acinus and showed shrunken nuclei and a swollen and pale appearance. (**D**–**F**) H&E staining of the human free SGs from the nasal wing and (**G**–**I**) from the lip and (**J**–**L**) from the external auditory canal (EAC). Human free SGs consisted of the central duct (asterisk) and several acini (arrow). Basal cells (arrowhead) underwent differentiation and finally formed matured cells (open arrowhead) in the nasal wing (**F**), the lip (**I**), and the external auditory canal (**L**), as the lipid droplets increase, and the nucleus shrinks. (**M**,**N**) H&E staining of the human hair-associated SGs. It consisted of the central duct (asterisk), several acini (arrow), the hair follicle (open arrow), and the hair. (**O**) The hair-associated SG acini. The basal cells (arrowhead) underwent differentiation and finally formed matured cells (open arrowhead), as the lipid droplets increase, and the nucleus shrinks. Scale bar 100 μm.

**Figure 2 ijms-25-03109-f002:**
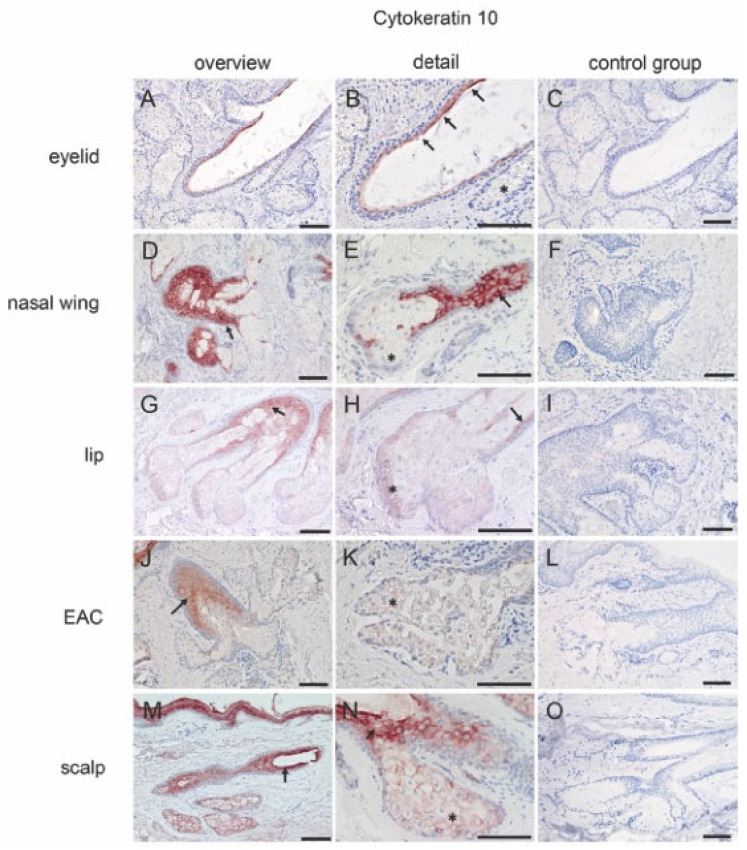
Immunohistochemical analysis of human tissue for CK10. The positive antibody reaction is shown by the red coloration. (**A**–**C**) MG. CK10 is expressed in the superficial layers of the central duct (arrow) but not in the acini (asterisk). (**D**–**F**) Free SG from nasal wing, (**G**–**I**) lip, and (**J**–**L**) EAC. Superficial layers of the central duct (arrow) show intensive reactivity, but weak reactivity in the free SG acini (asterisk). (**M**–**O**) Hair-associated SG. CK10 is expressed on both the superficial layers of the excretory duct (arrow) and SG acini (asterisk). No reactivity is visible in the negative control group. Scale bar 100 μm.

**Figure 3 ijms-25-03109-f003:**
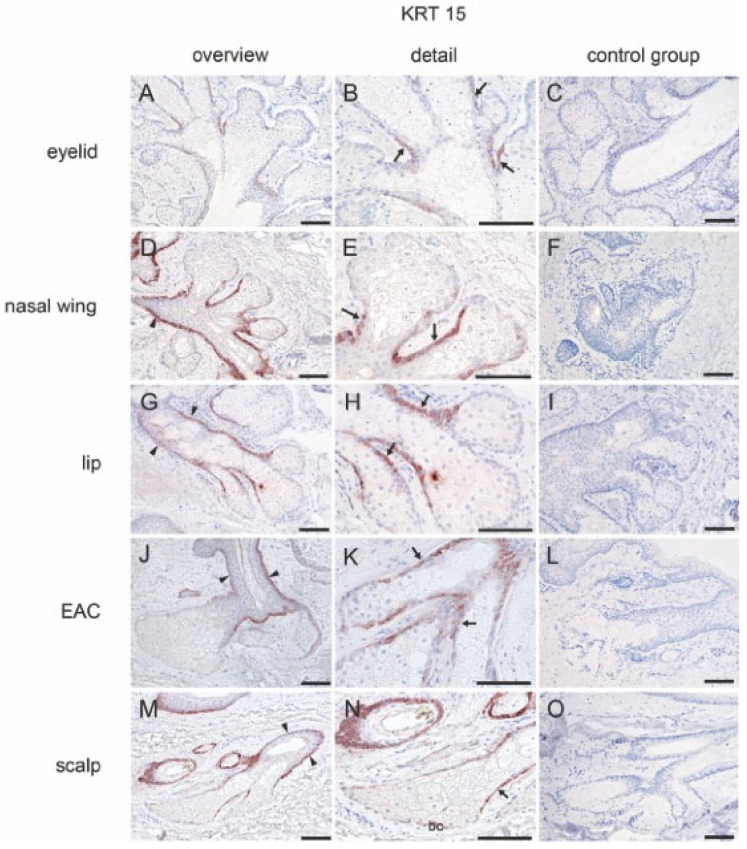
Immunohistochemical analysis of human tissue for KRT15. The positive antibody reaction is visible by the red coloration. (**A**–**C**) MG. KRT15 is expressed on the connecting ductules (arrow). (**D**–**F**) Free SG from nasal wing, (**G**–**I**) lip, and (**J**–**L**) EAC. KRT15 is expressed on the connecting ductules (arrow) and the basal layer of the central duct (arrowheads). (**M**–**O**) Hair-associated SG. KRT15 is expressed in the connecting ductules (arrow), the basal layer of the central duct (arrowheads), and also in the basal cells (bc). No reactivity was observed in the negative control group. Scale bar 100 μm.

**Figure 4 ijms-25-03109-f004:**
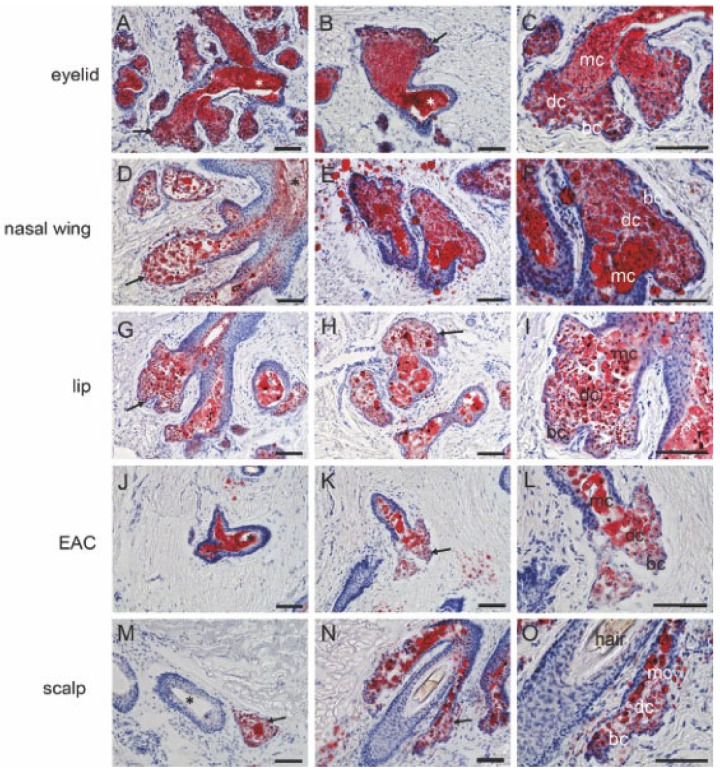
Lipid accumulation in human MGs, free SGs, and hair-associated SGs. (**A**–**C**) MG. Lipid droplets fill the entire central duct lumen of the MG (asterisks) and the acini (arrows). Here, lipids are present in the basal cells (bc), differentiating cells (dc), and mature cells (mc). The lipid droplets fill the entire mc, but they do not fill the entire bc and dc. (**D**–**F**) Free SG from nasal wings, (**G**–**I**) lips, and (**J**–**L**) external auditory canals. The lipid droplets are observed in bc, dc, and mc. The area in the mc is larger than in the bc and dc. The lipid droplets do not fill the entire free SG central duct (asterisk). (**M**–**O**) Hair-associated SG. The lipid droplets are observed in bc, dc, and mc, but not in the excretory duct (asterisk). Scale bar 100 μm.

**Figure 5 ijms-25-03109-f005:**
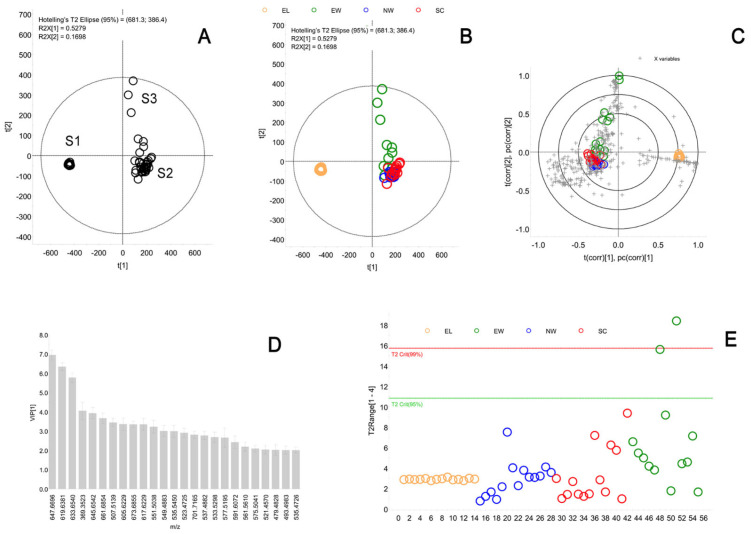
Comparative LC-MS analyses of the lipid composition of human meibum and of three types of sebum from different loci. The raw data were analyzed in the Progenesis QI software using its Principal Component Analysis (PCA) and Partial Least Squares Discriminant analysis (PLS-DA) algorithms. (**A**) A PCA scores plot of all four types of de-identified study samples. Three main groups of samples emerged. (**B**) The same plot as in Panel A after the four groups of samples were color-coded. EL—meibum; EW—ear sebum; NW—nasal sebum; SC—scalp sebum. The unbiased, unsupervised PCA approach revealed that meibum samples were highly different from other types of studied samples. Simultaneously, scalp sebum and nasal sebum were found to be almost identical, while ear sebum showed the highest variability among the study samples, with some of the samples being similar to ear and nasal sebum specimens, while the rest being different. (**C**) A supervised PLS-DA analysis of the same samples produced the same type of patterning and accentuated similarities between the SC and NW samples, and a vastly different lipidome of meibum. (**D**) Variable Importance plots were used to identify the lipids with the highest impact on the separation of the sample groups. Among these, lipids, wax esters, cholesteryl esters, and free cholesterol contributed the most to the separation of the study samples. (**E**) A Hotelling’s T^2^ range plot demonstrated that only one sample from the EW group (above the T2 Crit (99%) red line in the graph) might be considered a serious outlier while the rest of the samples are well explained by the model.

**Figure 6 ijms-25-03109-f006:**
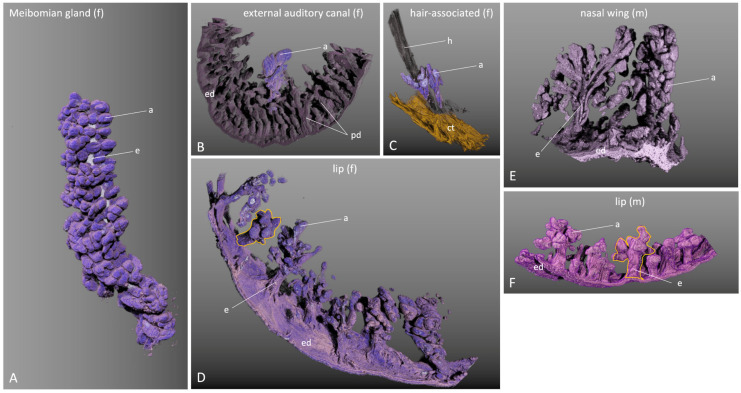
Three-dimensionally reconstructed sebaceous glands. (**A**) Illustration of a meibomian gland. The glandular acini (a) are colored purple. The excretory duct (e) shimmers white. This is the meibomian gland of an upper eyelid; the gland is oriented as it is embedded in the eyelid. (**B**) Sebaceous gland beneath the papillary dermis (pd) and epidermis (ed) from the external auditory canal. The specimen has contracted after dissection, so the surface with the epidermis (ed) appears to be curved the wrong way. The sebaceous gland is stained purple. Its glandular acini (a) appear “fused”. (**C**) Illustration of a hair shaft (h) around the lower end of which a sebaceous gland is situated (purple). The a = glandular acini points to an acinus (also colored purple) which already belongs to another hair, but which is not shown in the figure. Thus, two hair-associated sebaceous glands lie exactly next to each other. Some connective tissue (ct) is colored yellow for better orientation. (**D**) Illustration of sebaceous glands of the lip. A single sebaceous gland is circled in orange. A total of eight sebaceous glands of different sizes are shown. The excretory ducts (e) run toward the epidermis (ed), which is virtually seen from the inside. (**E**) Illustration of two sebaceous glands of the nostril. In the left gland, the main excretory duct system is “cut open” (e) and one looks into the branched larger ducts. (**F**) Illustration of sebaceous glands in the lip. A single gland is circled in orange. A total of seven sebaceous glands are visible, with their excretory ducts (e) opening onto the upper surface of the epidermis (ed). f = female, m = male, a = acinus, e = excretory duct, pd = papillary dermis, ed = epidermis, ct = connective tissue.

**Table 1 ijms-25-03109-t001:** Results of the immunohistochemical analysis of biomarkers within the different human glands. MG—meibomian gland; SG—sebaceous gland; EAC—external auditory canal; CK—cytokeratin; KRT—keratin; Dsg—desmoglein; Dsc—desmocollin; Dp—desmoplakin; Pg—plakoglobin; sl—superficial layer; bl—basal layer; bc—basal cells; mc—mature cells; cd—collecting ductule.

Markers	MGs				Free SGs												Hair-Associated SGs			
	Eyelid				Nasal Wing				Lip				EAC				Scalp			
	Duct		bc	mc	Duct		bc	mc	Duct		bc	mc	Duct		bc	mc	Duct		bc	mc
	sl	bl			sl	bl			sl	bl			sl	bl			sl	bl		
cytokeratins																				
CK1	+	−	−	−	+	−	−	−	+	−	−	−	+	−	−	−	+	−	+	+
CK8	−	−	+	−	−	−	+	−	−	−	+	−	−	−	+	−	−	−	+	−
CK10	+	−	−	−	+	−	+	+	+	−	+	+	+	−	+	+	+	−	+	+
CK14	−	+	+	−	−	+	+	+	−	+	+	+	−	+	+	+	−	+	+	+
stem cell marker	ductule	cd	bc	mc	ductule	cd	bc	mc	ductule	cd	bc	mc	ductule	cd	bc	mc	ductule	cd	bc	mc
KRT15	+	−	−	−	+	+	−	−	+	+	−	−	+	+	−	−	+	+	+	−
N-cadherin	−	−	+	−	−	−	+	−	−	−	+	−	−	−	+	−	−	−	+	−
cell–cell contact marker																				
Dsg1	+	+	+	+/−	+	+	+	+	+	+	+	+	+	+	+	+	+	+	+	+
Dsc3	+	+	+	+	+	+	+	+	+	+	+	+	+	+	+	+	+	+	+	+
Dp	+	+	+	+	+	+	+	+	+	+	+	+	+	+	+	+	+	+	+	+
Pg	+	+	+	+	+	+	+	+	+	+	+	+	+	+	+	+	+	+	+	+
E-cadherin	+		+	+	+	+	+	+	+	+	+	+	+	+	+	+	+	+	+	+
Claudin5	−	−	−	−	−	−	−	−	−	−	−	−	−	−	−	−	−	−	−	−

**Table 2 ijms-25-03109-t002:** Antibodies used.

Antibody	Species	Clonality	Company	Product No.	Dilution
Primary antibodies					
CK1	Mouse	Monoclonal	Santa Cruz	sc-376224	1:25
CK8	Rabbit	Monoclonal	Abcam	ab53280	1:250
CK10	Mouse	Monoclonal	Santa Cruz	sc-53252	1:250
CK14	Mouse	Monoclonal	Novocastra	NCL-LL002	1:20
KRT15	Mouse	Monoclonal	Abcam	ab80522	1:50
N-cadherin	Rabbit	Polyclonal	Abcam	ab225719	1:50
E-cadherin	Rabbit	Monoclonal	Abcam	ab40772	1:250
Dsg1	Mouse	Monoclonal	Progen	651111	1:20
Pg	Mouse	Monoclonal	Progen	61005	1:5
Dp	Rabbit	Polyclonal	St. John’s laboratory	ABIN2736659	1:100
Dsc3	Rabbit	Polyclonal	Abcam	ab190118	1:100
Claudin 5	Rabbit	Monoclonal	Thermo Fisher	JM11-22	1:50
Secondary antibodies					
Goat anti-rabbit biotinylated IgG		Polyclonal	Dako	E0432	1:200
Goat anti-rabbit biotinylated IgG		Polyclonal	Invitrogen	31820	1:200
Goat anti-mouse biotinylated IgG		Polyclonal	Dako	E0433	1:200

CK—cytokeratin; KRT—keratin; Dsg—desmoglein; Dsc—desmocollin; Dp—desmoplakin; Pg—plakoglobin; IgG—immunoglobulin G.

## Data Availability

The data presented in this study are available on request from the corresponding author (F.P.).

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
