# Peer review of "Comparative Characterization of Human Meibomian Glands, Free Sebaceous Glands, and Hair-Associated Sebaceous Glands Based on Biomarkers, Analysis of Secretion Composition, and Gland Morphology"

_ijms, 2024, doi:10.3390/ijms25063109_

Round 1

Reviewer 1 Report

Comments and Suggestions for Authors

Translator        

The manuscript by Liu et al. nicely describes a variety of human gland types, a study that will help researchers to better understand human gland morphology etc. in upcoming studies.

While the results are clearly presented, I would suggest the following:

1. A better resolution for the submitted figures

2.  Magnifications from the stainings to get a better idea of dye distribution and epithelial structure.

Author Response

Reviewer #1: The manuscript by Liu et al. nicely describes a variety of human gland types, a study that will help researchers to better understand human gland morphology etc. in upcoming studies.

While the results are clearly presented, I would suggest the following:

  1. A better resolution for the submitted figures
  2. Magnifications from the stainings to get a better idea of dye distribution and epithelial structure.

Answer: Following the suggestions of reviewer 1, we have improved the resolution of all images. An enlargement of the reaction result of the staining in all immunohistochemical images is shown in the middle row (named “detail”) to better assess the reactions at the epithelial level.

Reviewer 2 Report

Comments and Suggestions for Authors

In this manuscript, authors have characterized the presence of biomarkers in meibomian glands in comparison to free sebaceous glands and hair-associated sebaceous glands including comparison of their three-dimensional tissue structure. While lipidomic analysis is done it is not presented in detail in the current manuscript.

Major concerns:

Current work is limited to a healthy control group. Further analysis could have been done by recruiting study subjects with dysfunctional meibomian glands as in DED cases to examine the differential expression of biomarkers and secretion composition, if any. Also, the study is conducted on a small sample size which greatly undermines the conclusions made. Detailed lipidomic analysis are not presented citing space constraints.

Minor concerns:

Section 2.2: line 133- titled “Immunohistochemical localization of biomarkers in MG, free MG.....” Is it free MG or free SG?

Abbreviations should be elaborated in the legends itself. Example: sl and bl in Table 1 legend.

Comments on the Quality of English Language

 Minor editing of English language required.

Author Response

Reviewer #2:

In this manuscript, authors have characterized the presence of biomarkers in meibomian glands in comparison to free sebaceous glands and hair-associated sebaceous glands including comparison of their three-dimensional tissue structure. While lipidomic analysis is done it is not presented in detail in the current manuscript.

  1. Major concerns:

Current work is limited to a healthy control group. Further analysis could have been done by recruiting study subjects with dysfunctional meibomian glands as in DED cases to examine the differential expression of biomarkers and secretion composition, if any. Also, the study is conducted on a small sample size which greatly undermines the conclusions made. Detailed lipidomic analysis are not presented citing space constraints.

Answer: We agree in principle with the points raised by the reviewer. However, we do not consider the examination to be "limited" by the fact that only healthy samples were analyzed. We also not investigated a "control group" as the reviewer formulates it, but the “central investigation group” for this study. The examination was carried out in an institute of anatomy. Pathological changes can only be deduced and understood by examining first the healthy, physiological situation what was done here.

The reviewer is right, of course further analyses could have been performed, such as with dysfunctional meibomian glands as in DED cases, to understand the differential expression of biomarkers and secretion compositions. Such analyses are also planned for the future. However, we are not ophthalmologists and therefore do not have direct access to patients who would have to be extensively examined in advance in order to generate appropriate samples. These examinations would also only be possible with regard to the lipid analysis, but not possible with regard to the analysis of biomarkers and the morphological examinations we carried out, as it would be extremely difficult to classify body donors with MGD, in which further tissue samples with sebaceous glands are then taken from the corresponding body localizations for comparison with the meibomian glands and serial sections are produced for 3D reconstruction.

In our opinion, the lipid section cannot go beyond what was presented in the original manuscript. We have added a short paragraph to formally address the reviewer's comments in the Conclusions section (section 5, lines 751-756), but a full-scale structural and quantitative analysis of all types of secretions studied in the paper is a huge undertaking and it will not fit this manuscript as it will make the manuscript twice as long with twice as many figures. In fact, we plan to prepare a separate manuscript solely about the lipidomic analyses of these secretions in the near future.

We do not wish to offend the reviewer in any way but would like to emphasize that in our opinion his major concerns do not do justice to the value of the manuscript. We agree that the study "only" on samples from 4 body donors is not extensive at first glance, which we had already formulated as a study limitation. Nevertheless, on the other hand, it must be emphasized that all investigations were carried out on human tissue (not mice or other animals) from body donors and that for the first time a comparison of meibomian glands and sebaceous glands of different localization from the same people was made. If it were so easy to obtain samples, such a study would have been carried out long ago. For the biomarkers and morphological analysis, a total of 192 samples were taken (from the 4 body donors), all of which were embedded and then sectioned. Serial sections were prepared from 96 sample blocks, approx. 250 from each block, in order to create a 3D reconstruction of the glands. The biomarker analysis was carried out on the other 96 samples using 12 antibodies each. Not every immune reaction works immediately. Anyone familiar with serial sections, 3D reconstruction and immunohistochemistry will know how much work is involved. The results lead to numerous new insights despite the low initial n - number:

  1. The 3D reconstructions lead to a clearer understanding of gland morphology and they also show that the number of glandular acini in a meibomian gland is many times higher than previously reported in the literature.
  2. Our findings show numerous similarities between meibomian glands and sebaceous glands with regard to the biomarkers analyzed, but they also reveal differences that were previously unknown and are of interest for follow-up studies. The biomarker analyses also show differences between free and hair-associated lacrimal glands and are therefore also of interest to scientists who are not working solely on the ocular surface.
  3. The lipid analysis is a first clear indication that the lipid profile of meibomian glands and other sebaceous glands differs significantly. This has not been clear in this form until now. These analyses also show that there are more or less differences between the other sebaceous gland entities.
  4. For anatomists, especially histologists, the examination shows that sebaceous glands (including meibomian glands) differ not only morphologically but also in their secretion composition, which must be included in the relevant textbooks.

To summarize, on the one hand we agree with the reviewer's comments, on the other hand the present study is a fully comprehensive study in its own right. The points noted by the reviewer are good suggestions for follow-up studies that should be carried out by us or other working groups in the future and are derived from the present study. We ask the reviewer to take the above points into consideration when making his or her final decision and very much hope that we can convince him/her with our arguments that this is a fully-fledged study with new insights.

  1. Minor concerns:

Section 2.2: line 133- titled “Immunohistochemical localization of biomarkers in MG, free MG.....” Is it free MG or free SG?

Abbreviations should be elaborated in the legends itself. Example: sl and bl in Table 1 legend.

Answer: We thank the reviewer for these hints. In section 2.2: line 133 with the title "Immunohistochemical localization of biomarkers in MG, free MG..." free MG was corrected to free SG. In addition, the abbreviations in the table legends have been added, which we had previously forgotten.

All changes made in the text are marked in red in the text.

Round 2

Reviewer 2 Report

Comments and Suggestions for Authors

Authors have satisfactorily addressed the comments.